# Inhibition of α-Glucosidase and Pancreatic Lipase Properties of *Mitragyna speciosa* (Korth.) Havil. (Kratom) Leaves

**DOI:** 10.3390/nu14193909

**Published:** 2022-09-21

**Authors:** Thanchanok Limcharoen, Phisit Pouyfung, Ngamrayu Ngamdokmai, Aruna Prasopthum, Aktsar Roskiana Ahmad, Wisdawati Wisdawati, Woraanong Prugsakij, Sakan Warinhomhoun

**Affiliations:** 1School of Medicine, Walailak University, Nakhon Si Thammarat 80160, Thailand; 2Center of Excellent in Marijuana, Hemp and Kratom, Walailak University, Nakhon Si Thammarat 80160, Thailand; 3School of Public Health, Walailak University, Nakhon Si Thammarat 80160, Thailand; 4School of Pharmacy, Walailak University, Nakhon Si Thammarat 80160, Thailand; 5Biomass and Oil Palm Center of Excellent, Walailak University, Nakhon Si Thammarat 80160, Thailand; 6Department of Pharmacognosy and Phytochemistry, Faculty of Pharmacy, Universitas of Muslim Indonesia, Makassar 90241, Indonesia; 7Department of Pharmaceutics and Industrial Pharmacy, Faculty of Pharmaceutical Sciences, Chulalongkorn University, Bangkok 10330, Thailand

**Keywords:** *Mitragyna speciosa*, kratom, α-glucosidase, pancreatic lipase, anti-diabetes mellitus

## Abstract

Kratom (*Mitragyna speciosa* (Korth.) Havil.) has been used to reduce blood sugar and lipid profiles in traditional medicine, and mitragynine is a major constituent in kratom leaves. Previous data on the blood sugar and lipid-altering effects of kratom are limited. In this study, phytochemical analyses of mitragynine, 7-hydroxymitragynine, quercetin, and rutin were performed in kratom extracts. The effects on α-glucosidase and pancreatic lipase activities were investigated in kratom extracts and mitragynine. The LC-MS/MS analysis showed that the mitragynine, quercetin, and rutin contents from kratom extracts were different. The ethanol extract exhibited the highest total phenolic content (TPC), total flavonoid content (TFC), and total alkaloid content (TAC). Additionally, compared to methanol and aqueous extracts, the ethanol extract showed the strongest inhibition activity against α-glucosidase and pancreatic lipase. Compared with the anti-diabetic agent acarbose, mitragynine showed the most potent α-glucosidase inhibition, with less potent activity of pancreatic lipase inhibition. Analysis of α-glucosidase and pancreatic lipase kinetics revealed that mitragynine inhibited noncompetitive and competitive effects, respectively. Combining mitragynine with acarbose resulted in a synergistic interaction with α-glucosidase inhibition. These results have established the potential of mitragynine from kratom as a herbal supplement for the treatment and prevention of diabetes mellitus.

## 1. Introduction

Obesity is a major risk factor that is closely related to various pathological conditions and chronic diseases worldwide and has become a major global public health problem. In 2022, the World Health Organization (WHO) reported that among adults aged 18 years and older, 39% were overweight and 13% were obese [1]. Recent studies revealed that the common underlying cause of both diabetes and obesity is related to insulin resistance, which occurs due to the stimulation of insulin production or a reduction in insulin receptors [2,3]. Common molecular targets for designing anti-obesity drugs are enzymes, which are accounted for virtually half of the small-molecule drugs available on the market. Because of their protein structures with several validated sites for drug interaction, enzymes are proved to be an appealing target for the discovery of novel therapeutic molecules [4]. Natural products are known to be remarkable sources for discovering possible therapeutic approaches for metabolic disorders, such as anti-diabetic and anti-obesity drugs. Hence, extensive experiments were made to investigate the effects of molecules from natural products through enzyme inhibition assay, particularly activities of α -Glucosidase and lipase [5,6,7].

α-glucosidase is a carbohydrate-hydrolyzing enzyme that is secreted from the intestinal chronic epithelium. These enzymes are essential enzymes of digestion that stimulate the breakdown of disaccharides and oligosaccharides into small, simple, and absorbable carbohydrates [8]. One of the most common alpha-glucosidase inhibitors (AGIs) drugs currently available is acarbose, which is proven to be efficient in blood glucose level stabilization. Moreover, acarbose was found to have activity against oxidative stress and endothelial dysfunction; therefore, it might lower the risk of developing cardiovascular diseases and help increase the lifespans of type-2 DM patients [9]. In addition, the dysfunctions of insulin-producing pancreatic β cells lose their functions due to the excessive accumulation of lipids in the pancreas [10,11]. Lipase is an enzyme produced by the exocrine portion of the pancreas and is released into the intestinal lumen to catalyze the hydrolytic breakdown of triacylglycerols in ingested fats, into free fatty acids and monoacylglycerols; thus, it is the most important enzyme in lipid digestion and is absorbed into circulation [12]. Orlistat, a lipase inhibitor that competes with dietary lipid molecules for enzymatic active sites, is an archetypic medication prescribed for obese patients in need of losing weight. Orlistat, with a therapeutic oral dose of 120 mg administered three times per day, accounts for nearly one-third of the reduction of dietary fat absorption via intestinal epithelium due to its inhibitory effect against lipase in the hydrolysis of triacylglycerol. This decrease in lipid absorption is shown to be without prominent effects on appetite [13]. However, long-term use of these drugs is often reported to cause some severe side effects, such as insomnia, headaches, hypoglycemia, weight gain, constipation, and renal damage [14,15]. Efforts have been directed toward discovering medicines from natural products due to their low costs, relative safety, probability of high compliance, and low incidences of undesirable side effects [16]. Based on previous studies, secondary metabolites of natural products, such as phenolics, alkaloids, terpenoids, flavonoids, and glycosides, have shown potent pancreatic lipase and α-glucosidase inhibitory activities [17,18,19,20,21,22]. The α-glucosidase inhibitory potential was also shown by plants in the Juglandaceae family [23]. However, it remains necessary to continue searching for more effective α-glucosidase and lipase inhibitors from traditional herbs.

*Mitragyna speciosa* (Korth.) Havil. or kratom (Rubiaceae) is an indigenous plant in Southeast Asia that grows naturally in several regions, including Thailand, Indonesia, Malaysia, Sumatra, Java, Bali, and Borneo (Figure 1) [24]. Kratom leaves have been used in local communities to treat pain, cough, fever, and diabetes; to enhance work performance; and are used as substitutes for illicit substances, mainly opioids. The leaves have been used as a traditional medicinal herb in southern Thailand, where it has become a culturally accepted stimulant drink similar to coffee and tea [25]. Previous studies revealed that *M. speciosa* contains a diverse group of secondary metabolites, such as indole alkaloids, flavonoids, triterpenoids, saponins, and glycosides [26]. More than forty of these compounds have been identified. Mitragynine is the most abundant compound available in the kratom preparation, with an estimated 2% by mass and up to two-thirds (66%) of total alkaloid content [27]. In addition, through in vitro and in vivo studies, kratom has been observed to exhibit various pharmacological properties, such as antioxidant, anti-inflammation, antibacterial, antiproliferative, and anti-analgesic properties [28,29]. Although previous studies have reported that kratom leaves exhibit activities that control diabetes and lipid profile [30,31,32,33,34,35], there is still a lack of scientific evidence related to the inhibitory effects and mechanism of enzymatic activities. Thus, in this research, we aimed to evaluate the α-glucosidase and pancreatic lipase inhibitory activities of kratom leaves. Additionally, the phytochemical profile, total alkaloid content, total phenolic content, and total flavonoid content of kratom extract were also investigated.

## 2. Materials and Methods

### 2.1. Plant Material

Fresh leaves of *M. speciosa* were collected from Thasala district, Nakhon Si Thammarat Province, Thailand in February 2022 (Figure 1). The leaves were identified by the Plant Varieties Protection office, Department of Agriculture, Thailand; the voucher specimen (BK083621) was deposited.

### 2.2. Preparation of Kratom Extracts

The leaves of kratom (1 kg) were dried in a hot air oven at 60 °C and ground to a coarse powder (600 g). The kratom powder was extracted with various organic solvents, including methanol (MeOH), ethanol (EtOH), and water, respectively for 24 h (three times). The extracts were filtered using a Whatman No. 1 filter paper. Each filtrate was concentrated to dryness in a rotary evaporator (Büchi Labortechnik, Germany) under reduced pressure and controlled temperature (40 °C) to give the final extracts including EtOH extract (52.8 g), MeOH extract (60.5 g), and aqueous extract (57.99 g), which were stored at 4 °C in a refrigerator until further use.

### 2.3. Determination of Total Phenolic Content, Total Flavonoid Content, and Total Alkaloid Content

#### 2.3.1. Total Phenolic Contents (TPC)

The total phenolic content was determined for individual extracts using the Folin–Ciocalteu method with some modifications [36]. Briefly, the 20 µL of extracts were mixed with 100 µL of tenfold diluted Folin–Ciocalteu reagent (Sigma-Aldrich, St Louis, MO, USA) and 80 µL of sodium bicarbonate (75 g/L). The mixture was incubated at room temperature for 1 h, and the absorbance was measured at 765 nm with a microplate reader (Thermo Scientific, Göteborg, Sweden). All samples were analyzed in triplicate. Gallic acid (4–30 µg/mL) was used as a positive control (Sigma-Aldrich, St Louis, MO, USA, lot number 099K0128). The results are expressed as milligrams of gallic acid equivalent (mg GAE/g).

#### 2.3.2. Total Flavonoid Content (TFC)

The total flavonoid content method was determined by using an aluminum chloride colorimetric assay with some modifications [37]. Briefly, 50 µL of samples were mixed with 10 µL of 10% aluminum chloride (Sigma-Aldrich, St Louis, MO, USA), 1 M sodium acetate (Sigma-Aldrich, St Louis, MO, USA), and 150 µL of 95% ethanol. The mixtures were further incubated in the dark at room temperature for 40 min. The absorbance was measured at 415 nm using a microplate reader (Thermo Scientific, Göteborg, Sweden). All samples were analyzed in triplicate. A solution of quercetin (Sigma-Aldrich, St Louis, MO, USA, lot number Q4951) in a range of 10–100 µg/mL was used to prepare a standard curve for determining the total flavonoid contents. The results are expressed as milligrams of quercetin equivalent (mg QE/g).

#### 2.3.3. Total Alkaloid Content (TAC)

Bromocresol green solution (1 × 10^−4^) was prepared by heating 69.8 mg bromocresol green (Sigma-Aldrich, St Louis, MO, USA) with 3 mL of 2N NaOH and 5 mL of distilled water until completely dissolved and the solution was diluted to 1000 mL with distilled water. A phosphate buffer solution (pH 4.7) was prepared by adjusting the pH of 2 M sodium phosphate (71.6 g Na_2_HPO_4_ in 1 L of distilled water) to 4.7 with 0.2 M citric acid (42.02 g citric acid in 1 L of distilled water). Atropine standard (Glentham Life Sciences, Ltd., Wiltshire, United Kingdom, lot number 756VTN) solution (20–140 µg/mL) was dissolved at 1 mg in 10 mL of distilled water [38]. All samples were analyzed in triplicate. The results are expressed as milligrams of atropine equivalent (mg ATR/g).

### 2.4. Liquid Chromatography Analysis of Kratom Extracts

UHPLC model Ultimate 3000 with an LC-MS/MS model Altis Plus (Thermo Fisher Scientific, MA, USA) was used. All samples were filtrated by a nylon filter with a pore size of 0.22 µm. The scanning range was from 100 to 1700 *m/z* for MS/MS in positive mode. Separation was achieved through Thermo Hypersil GOLD C-18 (2.1 × 100 mm, 1.9 µm). The gradient mobile phase was a mixture of solvent A: 0.1% formic acid (pH 2.99), and solvent B: acetonitrile, with a flow rate of 0.5 mL/min. The gradient program was set as 0–0.5 min, 25% B; 3–4 min, 80–100% B; and 4–6.5 min, 25% B with an injection volume of 10 µL. Standard mitragynine (0.1–1 µg/mL) (Lipomed, Inc., lot number 1610.1B0.2), quercetin (0.1–1 µg/mL) (Sigma-Aldrich, St Louis, MO, USA, lot number Q4951), 7-hydroxymitragynine (0.1–1 µg/mL) (ChromaDex, Inc., lot number 00008624-00377), and rutin (0.1–1 µg/mL) (Acros Organics, lot number A0355330) were eluted at 2.08, 1.20, 3.06, and 1.22 min, respectively. The mass scan mode was the positive multiple reaction monitoring mode. The precursor ion and product ion were *m/z* 399.20 → 174.10 for mitragynine, *m/z* 415.20 → 190.10 for 7-hydroxymitragynine, *m/z* 303.05 → 229.00 for quercetin, and *m/z* 611.16 → 303.10 for rutin.

### 2.5. Kinetic Study of α-Glucosidase Inhibition

The assay was performed as described previously with some modifications [39]. The kinetic parameters (Michaelis–Menten constant; *K_m_* and maximum velocity; *V_max_*) of α-glucosidase were determined by using *p*-nitrophenol-α-D-glucopyranoside (*p*NPG) as a substrate. Briefly, 0.1 unit of α-glucosidase (Sigma-Aldrich, St Louis, MO, USA) were incubated at 37 °C with 100 µL of total reaction. After 10 min, the reactions were started by adding 50 µL *p*NPG (Sigma-Aldrich, St Louis, MO, USA, lot number 3698310) ranging from 0 to 15 mM, and the *p*-nitrophenol product was measured at 405 nm using a microplate reader (Multiskan SkyHigh, Thermo Scientific, Göteborg, Sweden). The substrate concentration rates were plotted against *p*NPG concentrations and fitted to the Michaelis–Menten equation. For the inhibition assay, 100 µL containing 0.1 unit of α-glucosidase was incubated at 37 °C with acarbose (positive control) (Sigma-Aldrich, St Louis, MO, USA, Lot number SLCF5122) or kratom extracts at various concentrations (0–100 µg/mL).

The reactions were started by adding 50 µL of 0.3 mM *p*NPG (*K_m_*), and the *p*-nitrophenol product was measured at 405 nm using a microplate reader. The IC_50_ was determined by comparing the rate of each test inhibitor concentration to that of the vehicle control (ethanol), and the IC_50_ was calculated by plotting the remaining activity of each inhibitor concentration to the log test inhibitor concentration. The inhibition modes of pure compounds (mitragynine) and the positive control (acarbose) against the α-glucosidase enzyme were determined with at least four different concentrations of test inhibitors (mitragynine 0–0.4 mM; acarbose 0–3 mM). After co-incubation with each inhibitor for 5 min at 37 °C, the reactions were initiated by adding the *p*-NPG substrate (0–15 mM) [39]. The inhibition constant (*K**_i_*) values were determined from Lineweaver–Burk Plots.

### 2.6. Kinetic Study of Pancreatic Lipase Inhibition

The kinetic parameters (*K_m_* and *V_max_*) of pancreatic lipase were determined using 4-methylumbelliferyl oleate (4MUO) as the substrate (Sigma-Aldrich, St Louis, MO, USA, Lot number BCCF8781). Briefly, 0.5 units of lipase (Sigma-Aldrich, St Louis, MO, USA, Lot number SLCG8579) were incubated in 96-well plates with a total volume of 100 µL (15 mM Tris-HCl buffer, pH 8.0) at 37 °C for 10 min. The reactions were initiated by adding 50 µL of 4MUO ranging from 0 to 5 mM. The reactions were left to proceed for 10 min, and the rates of the reactions were measured by monitoring the increase in 4-methylumbelliferone (4MU), a fluorescent product (excitation, 355 nm; and emission, 460 nm). The rates at which substrate concentrations were plotted against 4MUO concentrations and fitted to the Michaelis–Menten equation. For the inhibition assay, 0.5 units of lipase were incubated with orlistat (Sigma-Aldrich, St Louis, MO, USA, Lot number 0000117290) or kratom extracts at various concentrations (0–100 µg/mL) in a 100 µL total reaction volume at 37 °C. After 10 min, the reactions were started by adding 0.35 mM 4MUO. The reaction rates were monitored by measuring the release of 4MU from 4MUO. Fluorescence from the release of 4MU was measured using a microplate reader (Synergy Mx, Agilent Technology, Santa Clara, USA) with excitation and emission wavelengths of 355 and 460 nm, respectively. The remaining activity of lipase was determined by comparing the rate of each test inhibitor concentration to the vehicle control (ethanol), and IC_50_ was calculated by plotting the remaining activity of each inhibitor concentration to the log test inhibitor concentration [39].

### 2.7. The Combination of Kratom Extracts and Acarbose Inhibited Enzymatic α-Glucosidase Activities

For the combination inhibition of α-glucosidase, 0.1 units of α-glucosidase was incubated with acarbose ranging from 0 to 100 µM in the presence and absence of ethanol extract (IC_50_; 16 µg/mL), methanol extract (IC_50_; 42 µg/mL), aqueous extract (IC_50_; 70 µg/mL), and mitragynine (IC_50_; 82 µg/mL) at 37 °C in a total volume of 100 µL of 15 mM Tris-HCl buffer, pH 8.0. After 10 min, the residual reactions were started by the addition of 50 µL of 0.3 mM *p*NPG. The reactions were left to proceed for 10 min. The absorption at 405 nm was then measured using a microplate reader, and IC_50_ values were calculated by GraphPad Prism version 9.3.1.

### 2.8. Statistical Analysis

All data were obtained from three dependent experiments and are expressed as the mean ± standard deviation (SD). Statistical analysis was performed using GraphPad Prism 9.3.1 (GraphPad Software, Inc., San Diego, CA, USA) with one-way ANOVA. Differences with * *p* < 0.05 were statically significant.

## 3. Results

### 3.1. TPC, TFC, and TAC of All Kratom Extracts

The effects of the extraction solvents (ethanol, methanol, and aqueous) on the TPC, TFC, and TAC of the kratom leaf extracts were evaluated, as shown in Table 1. The TPC was calculated from the regression equation of the calibration curve (*Y* = 0.0236x; *R*^2^ = 0.9995) and expressed as mg GAE/g of samples. The TFC was also reported as mg QE/g of samples (*Y* = 0.0059x; *R*^2^ = 0.9923). The content of alkaloids was measured in terms of atropine equivalents (*Y* = 0.0052x; *R*^2^ = 0.9980). The results showed that the ethanol extract exhibited a higher TPC (252.92 ± 1.15 mg GAE/g), TFC (26.07 ± 0.01 mg GAE/g), and TAC (88.04 ± 0.15 mg ATR/g) than those of the other kratom extracts. These results suggest that the ethanol extract is rich in phenolics, flavonoids, and alkaloids.

### 3.2. LC-MS/MS Analysis of Mitragynine, 7-hydroxymitragynine, Quercetin, and Rutin

LC-MS/MS is a confirmed hyphenated and accurate tool for rapid analysis, and it was used for the identification of a total of the mitragynine, 7-hydroxymitragynine, and two flavonoid compounds, which were identified by comparing their retention times and mass fragmentation patterns with data obtained from previous studies [26]. Then, each individual compound was quantified by comparing its peak area with the calibration curve obtained for the corresponding standard (Figure 2). Table 2 summarizes the bioactive phytochemical, mainly indole alkaloids and flavonoids. The presence and identification of these phytochemicals correlate with the reports published in a previous report [26].

The quantitative analysis of kratom leaf extracts is shown in Table 3. Mitragynine appeared to be the major alkaloid that was found in ethanol, followed by methanol and aqueous extracts, respectively. From this result, we can calculate that the percentages of mitragynine in TAC are 66% in methanol extract, 68% in ethanol extract, and 6.9% in aqueous extract. However, 7-hydroxymitragynine was observed at less than 1 mg/g. Additionally, quercetin and rutin were found to contain the highest amounts of ethanol compared to the other kratom extracts.

### 3.3. α-Glucosidase Inhibition Activity

The three different solvent extracts of kratom were evaluated for their α-glucosidase inhibitory activities (Table 4). Each kratom extract was initially treated at 0–100 µg/mL. The ethanol extract showed the strongest activity (IC_50_ 15.9 ± 1.34 µg/mL), followed by methanol extract (IC_50_ 42.12 ± 1.76 µg/mL) and aqueous extract (IC_50_ 69.48 ± 2.67 µg/mL), with a potency higher than that of the drug acarbose (IC_50_ 728.20 ± 7.01 µg/mL). However, compared to acarbose as a positive control, mitragynine has lower IC_50_ (81.68 ± 1.70 µg/mL) and, thus, has higher inhibitory activity (Figure 3A,B).

The inhibition kinetics of acarbose and kratom extracts against α-glucosidase were analyzed using the Lineweaver–Burk plots of the inverted values of velocity (1/V) versus the inverted values of substrate concentration (1/[S]), which are presented in Figure 4A–D. The drug acarbose showed the intersection of the lines on the ordinate, indicative of mixed-type inhibition. The results showed that *k_m_* 0.3 mM and *V_max_* (28 µmol/min/mg) were consistent with a previous report [39]. The *K_i_* of each mode was evaluated (Figure 4A,B). The intersection on the abscissa yielded a *K_i_* value of acarbose (0.28 mM), indicating mixed-type inhibition. Mitragynine was present at concentrations of 100, 200, and 300 µM with a *K_i_* value of 0.10 mM (Figure 4C,D). These results suggest that mitragynine is a noncompetitive inhibitor of this enzyme (Table 5).

### 3.4. Pancreatic Lipase Inhibition Activity

The kratom leaf extracts were subjected to pancreatic lipase inhibitory activity with 4MUO as the substrate evaluation. Each kratom extract was initially treated at 0–100 µg/mL (Table 4). The results revealed that the ethanol extract exhibited strong inhibition of lipase (IC_50_ 14.15 ± 1.71 µg/mL), followed by the methanol extract (IC_50_ 28.38 ± 2.34 µg/mL) and aqueous extract (IC_50_ 41.43 ± 3.32 µg/mL), but the potency was lower than that of orlistat (IC_50_ 0.42 ± 0.05 µg/mL), which was a positive control (Figure 3C,D). In addition, we found that the IC_50_ value of mitragynine was 30-fold higher than that of orlistat. In experiments on the kinetics parameter of lipase with 4MUO as the substrate, the *K_m_* and *V_max_* values were determined to be 0.45 ± 0.01 mM and 58 nmol/min/mg, respectively. The *K_i_* values of orlistat and mitragynine were determined. The intersection on the abscissa yielded a *K_i_* value of orlistat (0.24 ± 0.03 µM) and mitragynine (14.94 ± 0.29 µM). These observations suggested that both orlistat and mitragynine were competitive inhibitors of the lipase enzyme (Table 5), but mitragynine was shown to have weaker inhibition than that of orlistat (Figure 4E–H).

### 3.5. The Combined Inhibitory Effect of Kratom Extracts and Mitragynine on α-Glucosidase Activities

The mode of inhibition between acarbose and mitragynine on the α-glucosidase enzyme was as described earlier. We hypothesized that acarbose and mitragynine might have synergist effects. The inhibitory effects of kratom and its combination with α-glucosidase are shown in Figure 5. The concentration of acarbose in the range of 0–15 mM was combined with the IC_50_ values of kratom extracts and mitragynine. The results showed that the IC_50_ of kratom extracts combined with acarbose was lowered almost two-fold compared with that of acarbose alone. Interestingly, combination treatments with mitragynine at the concentrations of 200 and 300 µM decreased IC_50_ values compared with that of mixed kratom extracts. However, 100 µM of mitragynine showed less potency than that of kratom extracts. Thus, the results indicated that mitragynine in kratom is a major constituent for reducing blood glucose as well as improving the efficiency of acarbose, which is a reference standard for glucose-lowering drug.

## 4. Discussion

*M. speciosa* or kratom contains more than 40 identified bioactive compounds including indole alkaloids, flavonoids, triterpenoids, saponins, and glycosides. These compounds have been described to have antioxidant, anti-inflammatory, antibacterial, antiproliferative, and analgesic activities by several studies [26,28,29,40]. In this study, the presence of phenolic, flavonoid, and alkaloid compounds in our kratom extracts were confirmed by TPC, TFC, and TAC, respectively (Table 1). The four major kratom extract constituents identified by LC-MS/MS included mitragynine, 7-hydroxymitragynine, quercetin, and rutin (Table 2), which corresponded to the previous studies [26,27]. The amounts of these four compounds appeared to be different depending on the extraction solvent used (Table 3). Ethanol, the most common extraction solvent used in maceration, could extract the highest amounts of all types of compounds, while water extracted the least amount of all the compounds regardless of the polarities of the compounds. This result could imply that the polarities of the extracting solvents directly affected the yield of extraction following the like–dissolve–like principle [41]. Additionally, these data suggest that the water-based preparation of kratom leaves as a tea or stimulant drink for use as an alternative medicine for pain relief and diabetes in different cultures, including the Thai culture [25], may not be effective, as water cannot extract many bioactive ingredients from kratom leaves.

Bioactive constituents of kratom leaves, including mitragynine, are already well-known for their analgesic activities, primarily via activation of μ-opioid receptors, in which 7-hydroxymitragynine has 16-fold higher analgesic activity than that of mitragynine. [42]. Increasing evidence indicates that kratom extract could be used to treat metabolic syndrome, i.e., controlling blood glucose and lipid profiles [30,31,32,33,34,35], even though information about the underlying molecular mechanisms or targeted molecules, in which bioactive compounds in kratom leaves exhibit these actions, is limited. There has been only one previous study that reported the potential of kratom leaf extracts in glucose-lowering effects, in which the methanolic extract of kratom leaves and the major constituent mitragynine promoted in vitro glucose uptake to muscle cells via glucose transporter-1 (GLUT-1) [31]. Thus, this study evaluated whether kratom extracts and the major constituent mitragynine could inhibit the enzymatic activities of α-glucosidase and pancreatic lipase, which are two of the most common molecular targets of anti-diabetic agents (i.e., acarbose) and anti-obesity agents (i.e., orlistat), respectively [5,6,7]. Inhibiting α-glucosidase results in fewer transformations of the oligosaccharides and disaccharides to glucose, which plays an important role in controlling the postprandial blood glucose levels of diabetics and keeping the blood glucose levels normal by delaying the digestion of carbohydrates and diminishing the absorption of monosaccharides [43], meanwhile, the suppression of pancreatic lipase activity reduces the breakdown of dietary triglycerides into free fatty acids and glycerol and, thus, helps lower blood triglyceride levels [12].

Among the three different kratom leaf extracts, ethanolic extract showed the strongest inhibitory effects toward both α-glucosidase and pancreatic lipase, followed by the methanolic and aqueous extracts (Table 4). It appeared that these trends of inhibition also corresponded to the amounts of the identified four major bioactive compounds that were identified in kratom extracts, which were highest in ethanolic extracts, followed by methanolic and aqueous extracts. Therefore, our results suggested that the highest inhibitory activities of the ethanolic extract were attributed to the presence of the four main compounds, mitragynine, 7-hydroxymitragynine, quercetin, and rutin, which were present in the highest amounts compared to the methanolic and aqueous extracts. Interestingly, it should be noted that all the kratom extracts outperformed the anti-diabetic agent acarbose, which is well known as an α-glucosidase inhibitor, to inhibit α-glucosidase activity [9]. Our discovery could primarily imply that kratom extracts could be used for lowering blood glucose levels by inhibiting the α-glucosidase enzyme. However, this was not a similar case for the inhibition, in which the kratom extracts appeared to inhibit pancreatic lipase activity 30-fold compared to inhibition by the well-known drug inhibitor orlistat [13]. These compounds in the kratom extracts could act synergistically for the activation or downregulation of some key pathways [44].

To further investigate whether the major constituent mitragynine mainly contributed to the inhibition of both α-glucosidase and pancreatic lipase, the purified mitragynine compound was also used in enzymatic assays and the determination of inhibition kinetics. We found that mitragynine exhibited an approximately 3.5-fold stronger inhibitory effect toward α-glucosidase than that of acarbose (Table 5). However, mitragynine appeared to exhibit weaker inhibition toward pancreatic lipase when compared to the known inhibitor drug orlistat judging by double-reciprocal plots (Figure 4A–H), mitragynine inhibited α-glucosidase in a noncompetitive mode with a *K_i_* value of 0.10 mM, whereas the known inhibitor drug acarbose inhibited α-glucosidase by a mixed-type inhibition mode with a *K_i_* value of 0.28 mM. In contrast, both mitragynine and the known inhibitor drug orlistat inhibited pancreatic lipase in a competitive mode, in which the *K_i_* of mitragynine was approximately 62.5-time higher than that of orlistat (*K_i_* = 14.94 ± 0.29 µM versus *K_i_* = 0.24 ± 0.03 µM, respectively). This finding suggested that mitragynine exhibited relatively weak inhibition toward pancreatic lipase and could not be effective for use in obesity management via lowering blood triglyceride levels. The orlistat molecule comprises several aliphatic chains that could span over the active site of the pancreatic lipase enzyme [45]. However, the relatively more rigid molecular structure of mitragynine may not fit well into the active site of pancreatic lipase, resulting in compromised inhibitory activities. Based on molecular docking, several previous studies have predicted that flavonoids with subclasses of flavones, flavanones, and chalcones could be potential candidate compounds for the effective inhibition of pancreatic lipase [46]. Further work will be conducted to investigate whether the other bioactive compounds in kratom leaves could exhibit anti-diabetic and anti-obesity properties.

It has been hypothesized that the two inhibitors with different modes of inhibition could contribute synergistically to each other (to the inhibition of α-glucosidase). Therefore, we investigated whether this postulation was correct by performing synergistic inhibition assays. When mitragynine was added to the enzymatic reactions in which acarbose was also present, a synergistic effect could be observed as the IC_50_ value of the compound mixture was lower than that of acarbose alone. Therefore, our results have reported for the first time the potential of kratom leaf extracts and the major constituent mitragynine for use as herbal medicinal therapies for diabetes. In addition, our information could primarily provide healthcare professionals with significant notes on the potential of the use of kratom in combination with the anti-diabetic agent acarbose for more effective control of blood glucose levels in patients with diabetes and can increase the knowledge [47] of using kratom in combination with bioactive substances and medicines.

## 5. Conclusions

Kratom leaves are known to be rich sources of alkaloids, flavonoids, and phenolic compounds, and our study found that the ethanolic extraction of kratom leaves produced the highest amount of these compounds compared with that of methanolic and aqueous extractions. The extracted compounds from kratom leaves also exhibited inhibitory activity against α-glucosidase and pancreatic lipase, and the highest inhibition was found from the ethanolic extract. Mitragynine, one of the major alkaloid constituents in kratom leaves, was found to have stronger inhibitory activity against α-glucosidase than that of the well-known anti-diabetic drug acarbose. We deduced that mitragynine is a main component for the inhibition of α-glucosidase in kratom leaves since the combination of acarbose with mitragynine showed a higher inhibitory effect than that of acarbose combined with kratom leaf extracts. In addition to the mentioned effect against α-glucosidase, the ethanol extracts of kratom leaves and mitragynine were revealed to possess repressive activity against pancreatic lipase. Our research suggests that kratom leaves, particularly mitragynine, have promising potential for use in therapeutic and protective applications in diabetic patients as herbal supplements in conjunction with standard pharmacological approaches.

## Figures and Tables

**Figure 1 nutrients-14-03909-f001:**
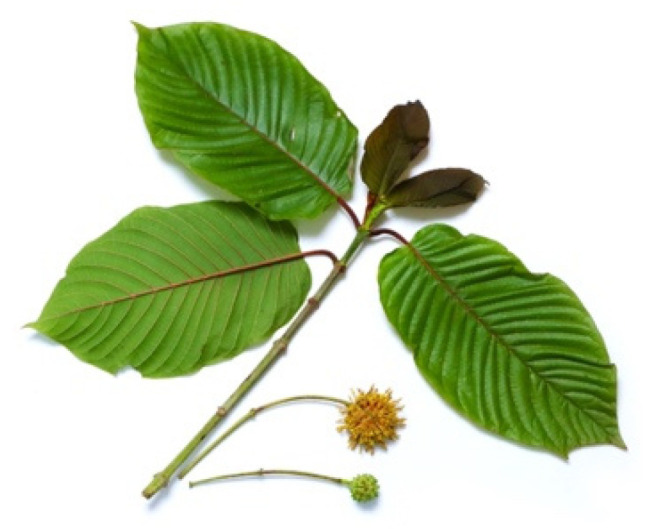
*Mitragyna speciosa* (Korth.) Havil. or Kratom.

**Figure 2 nutrients-14-03909-f002:**
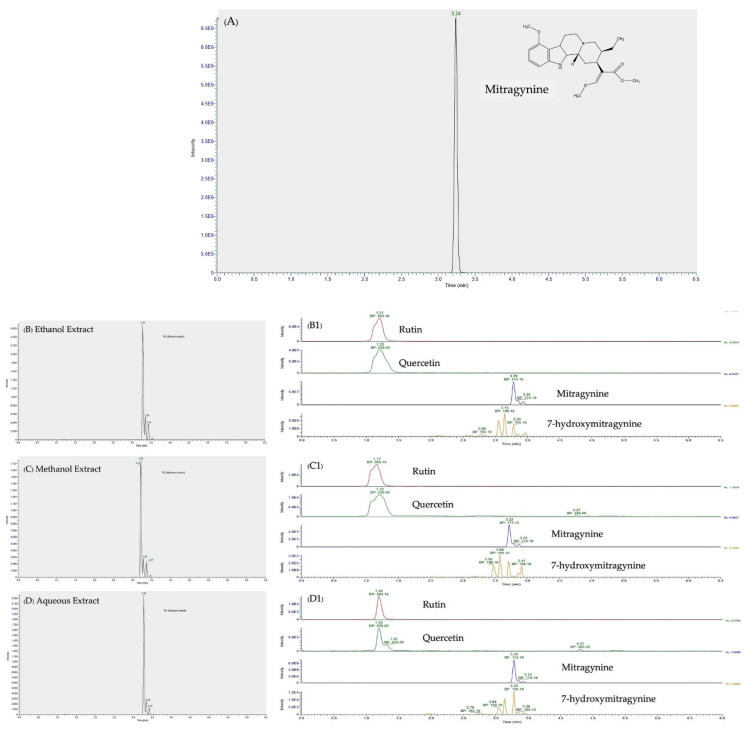
Total ion chromatograms (TIC) from LC-MS/MS of (**A**) the reference standard (mitragynine) concentration at 1 ug/mL, (**B**) ethanol extract, (**C**) methanol extract, and (**D**) aqueous extract from kratom leaves. The peaks of four major constituents were identified by comparison with the reference standards, their retention times, and mass fragmentation patterns (**B1**–**D1**) as rutin, quercetin, mitragynine, and 7-hydroxymitragynine.

**Figure 3 nutrients-14-03909-f003:**
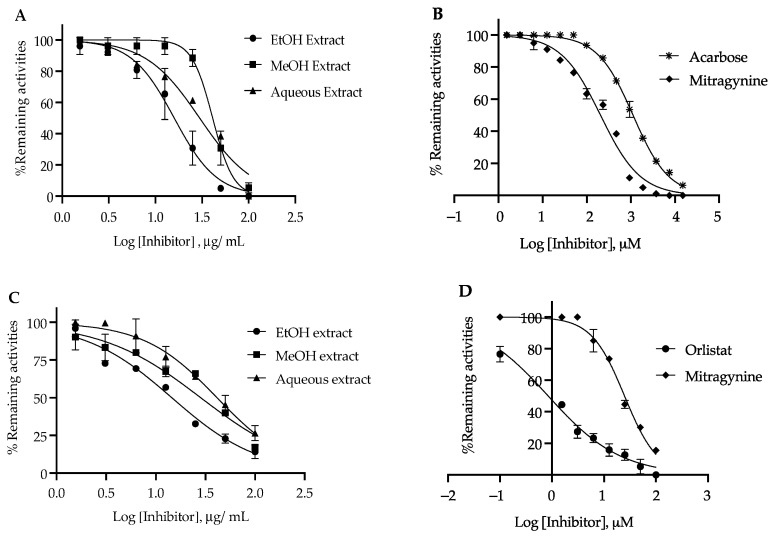
Inhibitory effect of the kratom extracts and mitragynine against α-glucosidase (**A**,**B**) and pancreatic lipase (**C**,**D**). Inhibition curves of acarbose (**B**) and orlistat (**D**) were used as positive controls.

**Figure 4 nutrients-14-03909-f004:**
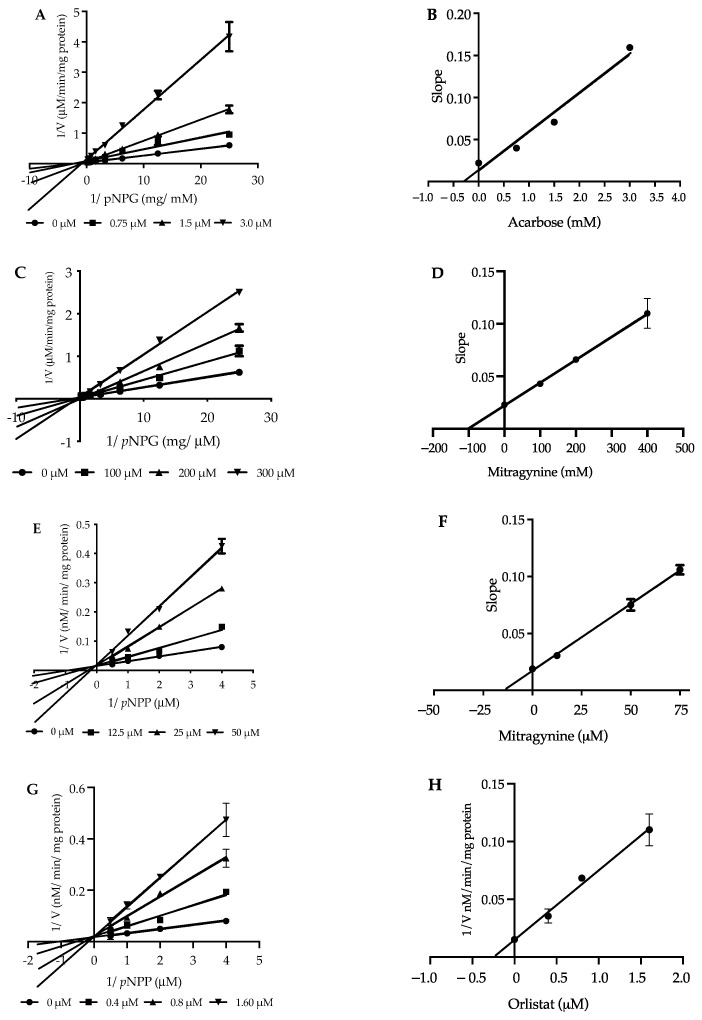
Lineweaver–Burk plots of α-glucosidase (**A**–**D**) and pancreatic lipase (**E**–**H**) in the presence and absence of kratom extracts, mitragynine, and positive controls (acarbose and orlistat).

**Figure 5 nutrients-14-03909-f005:**
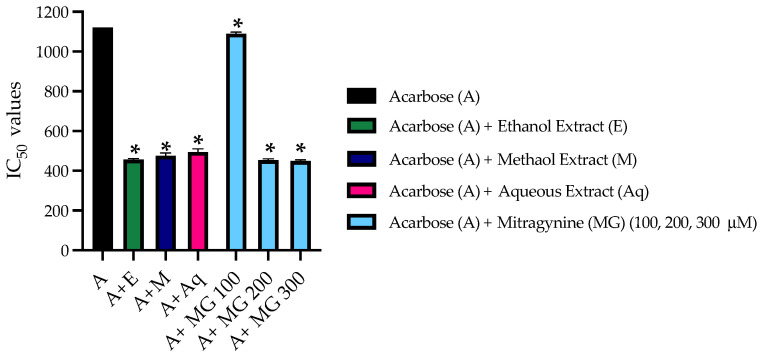
The IC_50_ values α-glucosidase of combinatorial kratom extracts and mitragynine with acarbose. The results are expressed as the means ± SDs, *n* = 3. * *p* < 0.05 compared to acarbose alone.

**Table 1 nutrients-14-03909-t001:** TPC (mg GAE/g extract), TFC (mg QE/g extract), and TAC (mg ATR/g extract) of kratom leaf extracts.

Samples	TPC (mg GAE/g) ± SD	TFC (mg QE/g) ± SD	TAC (mg ATR/g) ± SD
Ethanol	252.92 ± 1.15 *	26.07 ± 0.01 *	88.04 ± 0.15 *
Methanol	159.30 ± 2.01	13.15 ± 0.09	52.82 ± 0.85
Aqueous	130.58 ± 0.68	0.82 ± 0.02	5.61 ± 0.13

Results are expressed as means ± SDs, *n* = 3. * *p* < 0.05 compared to methanol and aqueous extracts of each group.

**Table 2 nutrients-14-03909-t002:** MS/MS data of compounds identified tentatively in kratom ethanol, methanol, and aqueous leaf extracts using UHPLC and LC-MS/MS.

Identification	Calculated *m/z* [M+H]^+^	Precursor ion Experimental *m/z* [M+H]^+^	Major Ion in MS/MS Spectra (Key Fragment Ions)	Ethanol RT, min	Methanol RT, min	Aqueous RT, min
Mitragynine	399.2278	399.20	174.10	3.284	3.224	3.224
7-hydroxymitragynine	415.2227	415.2	190.10	3.286	3.212	3.212
Rutin	611.1602	611.16	303.10	1.209	1.175	1.175
Quercetin	303.0508	303.05	229.00	1.217	1.223	1.223

**Table 3 nutrients-14-03909-t003:** Quantitative analysis of mitragynine, quercetin, and rutin of kratom extracts.

Compounds	Amount (mg/g) ± SD		
	Ethanol Extract	Methanol Extract	Aqueous Extract
Mitragynine	58.75 ± 0.21 *	35.87 ± 1.01	3.85 ± 0.17
Quercetin	19.10 ± 0.85 *	5.90 ± 0.14	1.28 ± 0.02
Rutin	11.36 ± 0.11 *	3.19 ± 0.22	1.22 ± 0.05

Results are expressed as means ± SDs, *n* = 3. * *p* < 0.05 compared to methanol and aqueous extracts of each group.

**Table 4 nutrients-14-03909-t004:** The IC_50_ values of kratom extracts and mitragynine for α-glucosidase and pancreatic lipase inhibitory activities.

Samples	α-Glucosidase		Pancreatic Lipase	
	IC_50_ (µM)	IC_50_ (µg/mL)	IC_50_ (µM)	IC_50_ (µg/mL)
Ethanol extract	-	15.90 ± 1.34 *	-	14.15 ± 1.71 *
Methanol extract	-	42.12 ± 1.76 *	-	28.38 ± 2.34 *
Aqueous extract	-	69.48 ± 2.67 *	-	41.43 ± 3.32 *
Mitragynine	205.04 ± 15.11 *	81.68 ± 1.70 *	24.9 ± 1.38 *	9.86 ± 0.45 *
Acarbose	1121.09 ± 67.01	728.20 ± 7.01	-	-
Orlistat	-	-	0.84 ± 0.10	0.42 ± 0.05

Results are expressed as means ± SDs, *n* = 3. * *p* < 0.05 compared to positive controls (acarbose and orlistat).

**Table 5 nutrients-14-03909-t005:** Kinetic parameters in α-glucosidase and pancreatic lipase in the presence of mitragynine and positive controls.

Inhibitors	α-Glucosidase		Pancreatic Lipase	
	K_i_ (mM)	Mode	K_i_ (µM)	Mode
Mitragynine	0.10	noncompetitive	14.94	competitive
Acarbose	0.28	mixed-type	-	-
Orlistat	-	-	0.24	competitive

## Data Availability

Data are contained within the article.

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
