# Peer review of "Inhibition of α-Glucosidase and Pancreatic Lipase Properties of Mitragyna speciosa (Korth.) Havil. (Kratom) Leaves"

_nutrients, 2022, doi:10.3390/nu14193909_

Round 1

Reviewer 1 Report

The paper provides a comprehensive analysis of the kratom (Mitragyna speciosa Korth) leaf extracts and the effects on α-glucosidase and pancreatic lipase activitiesThe subject is important and the manuscript is well structured and written. 

For the improvement of the manuscript, I have several suggestions:

Lines 30-32: please revise sentence (“compared with anti-diabetic agent acarbose” shows twice)

Lines 64-70: there are repeating ideas in the two sentences

Lines 80-82: α-Glucosidase inhibitory potential was also shown by plants in Juglandaceae family (doi: 10.3390/antiox10040607)

Lines 94-95: Eastlack et al. (26) was cited, however, please use your words 

Line 124: a citation should be given for this protocol (TPC)

Line 134: a citation should be given for this protocol (TFC)

Line 143: “per gram dry weight of the extract”?

Line 145: a citation should be given for this protocol (TAC)

Line 153: how were the results expressed?

Line 191: a citation should be given for this protocol

Line 239: “TPC, TFC, and TAC of kratom leaextracts.

Line 257: “... leaf extracts...”

Line 267: Table 4 should be moved after Line 272

Line 276: “mitragynine was less inhibited”?

Line 277: “extracts”

Line 302: Fig. 3 should be moved after Line 277; Fig. 3B “Mitragynine” 

Line 324: Fig. 4 should be moved after Line 280

Line 330: a new published review can be cited here (doi: 10.1002/hup.2805)

Line 331: “confirmed instead of were confirmed

Lines345-348: please revise sentence

Lines 371-374: because quercetin and rutin were mentioned, I suggest adding this idea - These compounds could act synergistically for the activation or down-regulation of some key pathways (doi: 10.3390/antiox11071412)

Lines379-381: please revise sentence

Line 415: you can add the following - “and can increase the knowledge (doi: 10.3389/fphar.2022.801855) of using kratom in combination with bioactive substances and medicines

Lines 428-429: “... have inhibitory effects against pancreatic lipase similar to those of α-Glucosidase.” Please rephrase

Author Response

Comments and Suggestions for Authors from reviewer 1

       The paper provides a comprehensive analysis of the kratom (Mitragyna speciosa Korth) leaf extracts and the effects on α-glucosidase and pancreatic lipase activities. The subject is important, and the manuscript is well structured and written. For the improvement of the manuscript, I have several suggestions:

Response : Thank you very much for your valuable suggestion. We have revised the manuscript according to the reviewer’s comments. The manuscript has been extensively edited as suggested.

  1. Lines 30-32: please revise sentence (“compared with anti-diabetic agent acarbose” shows twice)

Response : We apologize for this mistake. The sentence has been re-written as indicated with the yellow highlight in the revised manuscript (line 29-30).

Original sentences : Compared with the anti-diabetic agent acarbose, mitragynine showed the most potent α-glucosidase inhibition compared with anti-diabetic agent acarbose, with less potent activity of pancreatic lipase inhibition.

Revised sentences : Compared with the anti-diabetic agent acarbose, mitragynine showed the most potent α-glucosidase inhibition, with less potent activity of pancreatic lipase inhibition.

  1. Lines 64-70: there are repeating ideas in the two sentences

Response : We apologize for this mistake. The sentence has been re-written as indicated with the yellow highlight in the revised manuscript (line 63-66).

Original sentences : Pancreatic lipase is the key enzyme in lipid digestion and is responsible for the absorption of dietary fats through the breakdown of triacylglycerols into free fatty acids and monoacylglycerols in the intestinal lumen [12]. Lipase is an enzyme produced by exocrine portion of pancreas and released into intestinal lumen to catalyze hydrolytic breakdown of triacylglycerols in ingested fats into free fatty acids and monoacylglycerols, thus it is the most important enzyme in lipid digestion and absorption into circulation.

Revised sentences : Lipase is an enzyme produced by exocrine portion of pancreas and released into intestinal lumen to catalyze hydrolytic breakdown of triacylglycerols in ingested fats into free fatty acids and monoacylglycerols, thus it is the most important enzyme in lipid digestion and absorption into circulation [12].

  1. Lines 80-82: α-Glucosidase inhibitory potential was also shown by plants in Juglandaceae family (doi: 10.3390/antiox10040607)

Response : Thank you for your suggestion. We have added “α-glucosidase inhibitory potential was also shown by plants in Juglandaceae family ” as indicated with a yellow highlight in the revised manuscript (line 79-80)

  1. Lines 94-95: Eastlack et al. (26) was cited, however, please use your words

Response : We apologize for this mistake. The sentence has been re-written as indicated with the yellow highlight in the revised manuscript

Original sentences : The most prevalent compound is mitragynine, which accounts for approximately 2% of kratom preparations by mass, but up to 66% of the total alkaloid content (line 94-95).

Revised sentences : Mitragynine is the most abundant compound available in the kratom preparation, estimating 2% by mass and up to two-thirds (66%) of total alkaloid content (line 92-93).

  1. Line 124: a citation should be given for this protocol (TPC)

Response : Thank you for your comment. We have added citation as indicated with a yellow highlight in the revised manuscript (line 124)

  1. Line 134: a citation should be given for this protocol (TFC)

Response : Thank you for your comment. We have added citation as indicated with a yellow highlight in the revised manuscript (line 134)

  1. Line 143: “per gram dry weight of the extract”?

Response : We apologize for this mistake. The sentence has been re-written as indicated with the yellow highlight in the revised manuscript (line 130-131).

Original sentences : The results were expressed as milligrams of gallic acid equivalent per gram dry weight of the extract (mg GAE/g).                                                     Revised sentences : The results were expressed as milligrams of gallic acid equivalent (mg GAE/g).

  1. Line 145: a citation should be given for this protocol (TAC)

Response : Thank you for your comment. We have added citation as indicated with a yellow highlight in the revised manuscript (line 151).

  1. Line 153: how were the results expressed?

Response : We apologize for this mistake. The sentence has been added as indicated with the yellow highlight in the revised manuscript (line 171).

  1. Line 191: a citation should be given for this protocol

Response : Thank you for your comment. We have added citation as indicated with a yellow highlight in the revised manuscript (line 151-152).

  1. Line 239: “TPC, TFC, and TAC of kratom leaf extracts.”

Response : We apologize for this mistake. We have replaced “TPC, TFC, and TAC on the leaves of kratom” with “TPC, TFC, and TAC of the kratom leaf extracts” as indicated with yellow highlight in the revised manuscript (line 230).

  1. Line 257: “... leaf extracts...”

Response : We apologize for this mistake. We have replaced “kratom leaf extract” with “kratom leaf extracts” as indicated with yellow highlight in the revised manuscript (line 266)

  1. Line 267: Table 4 should be moved after Line 272

Response : Thank you for your suggestion. We have removed Table 4 to line 285 as indicated with yellow highlight in the revised manuscript.

  1. Line 276: “mitragynine was less inhibited”?

Response : We apologize for this mistake. We have replaced “Each of  kratom  extract” with “Each of kratom extracts” as indicated with yellow highlight in the revised manuscript (line 281-283)

Original sentences : However, compared to the positive control, mitragynine was inhibited lower than (IC50 81.68 ± 1.70 µg/mL) (Figure 3A-2B).

Revised sentences : However, compared to acarbose as a positive control, mitragynine has lower IC50 (81.68 ± 1.70 µg/mL) and thus has higher inhibitory activity (Figure 3A-2B).

  1. Line 277: “extracts”

Response : We apologize for this mistake. We have replaced “Each of  kratom  extract” with “Each of kratom extracts” as indicated with yellow highlight in the revised manuscript (line 278)

  1. Line 302: Fig. 3 should be moved after Line 277; Fig. 3B “Mitragynine”

Response : Thank you for your suggestion. We have removed Figure 3 to line 287 as indicated with yellow highlight in the revised manuscript.

  1. Line 324: Fig. 4 should be moved after Line 280

Response : Thank you for your suggestion. We have removed Figure 4 to line 300-302 as indicated with yellow highlight in the revised manuscript.

  1. Line 330: a new published review can be cited here (doi: 10.1002/hup.2805)

Response : Thank you for your comment. We have added citation as indicated with a yellow highlight in the revised manuscript (line 341)

  1. Line 331: “confirmed” instead of “were confirmed”

Response : We apologize for this mistake. We have replaced “were confirmed” with “confirmed” as indicated with yellow highlight in the revised manuscript (line 342)

  1. Lines345-348: please revise sentence

Response : We apologize for this mistake. The sentence has been re-written as indicated with the yellow highlight in the revised manuscript.

Original sentences : Our kratom extracts confirmed the presence of phenolic, flavonoid and alkaloid compounds by TPC, TFC and TAC, respectively (line 345-348).

Revised sentences : In this study, the presence of phenolic, flavonoid and alkaloid compounds in our kratom extracts were confirmed by TPC, TFC and TAC, respectively (line 341-343).

  1. Lines 371-374: because quercetin and rutin were mentioned, I suggest adding this idea - These compounds could act synergistically for the activation or down-regulation of some key pathways (doi: 10.3390/antiox11071412)

Response : : Thank you for your suggestion. We have added this sentence in the revise manuscript  (line 390-391).

  1. Lines379-381: please revise sentence

Response : We apologize for this mistake. The sentence has been re-written as indicated with the yellow highlight in the revised manuscript (line 379-381).

Original sentences : while the inhibition of pancreatic lipase leads to the inhibition of dietary triglyceride digestion into free fatty acids and glycerol and helps to reduce blood triglyceride levels (line 379-381).

Revised sentences : meanwhile suppression of pancreatic lipase activity reduces breakdown of dietary triglyceride into free fatty acids and glycerol, thus helps lowering blood triglyceride levels (line 372-374).

  1. Line 415: you can add the following - “and can increase the knowledge (doi: 10.3389/fphar.2022.801855) of using kratom in combination with bioactive substances and medicines”

Response : : Thank you for your suggestion. We have added this sentence in the revised manuscript  (line 425-426).

  1. Lines 428-429: “... have inhibitory effects against pancreatic lipase similar to those of α-Glucosidase.” Please rephrase

Response : We apologize for this mistake. The sentence has been re-written as indicated with the yellow highlight in the revised manuscript.

Original sentences : In addition, the ethanolic extracts of kratom leaves and mitragynine were found to have inhibitory effects against pancreatic lipase similar to those of α-Glucosidase (line 443-444).

Revised sentences : In addition to mentioned effect against α-glucosidase, both the ethanol extract of kratom leaves and mitragynine were revealed to possess repressive activity against pancreatic lipase (line 438-440).

Reviewer 2 Report

The objective of the current study aimed to evaluate for their α-glucosidase and pancreatic lipase inhibitory activities of kratom leaves. The results show that mitragynine from kratom could be used as an herbal supplement for the treatment and prevention of diabetes mellitus. However, some modifications should be revised before being accepted for publication.

1 Line 270 has an extra ". " after the" α- ".

2 Please add the doi number at the very end of the reference.

3 Please indicate in Table1, Table3, Table4 and Table5 whether there are significant differences between each group. After the analysis, the data were more convincing.

4 In line 257, there are multiple spaces before "LC-MS/MS".

5 "Tris-CL" in line 218 is changed to "Tris-HCL".

6 Please unify the upper and lower case of "α-glucosidase" in the whole text. For example, "α-glucosidase in line 37 and 390 is used in the sentence, but" α-Glucosidase "is used in other sentences in the text".

7 Figure4 and Figure5 are in reverse order, please correct them.

8 In this article, only the extracted ion current diagram of Mitragynine is presented. It is suggested that the authors add a total ion current diagram for liquid identification and give a brief description of compound identification.

Author Response

Comments and Suggestions for Authors from reviewer 2

    The objective of the current study aimed to evaluate for their α-glucosidase and pancreatic lipase inhibitory activities of kratom leaves. The results show that mitragynine from kratom could be used as an herbal supplement for the treatment and prevention of diabetes mellitus. However, some modifications should be revised before being accepted for publication.

Response : Thank you very much for your valuable suggestion. We have revised the manuscript according to the reviewer’s comments. The manuscript has been extensively edited as suggested.

1 Line 270 has an extra ". " after the" α- ".

Response : We apologize for this mistake. We have replaced “α.-Glucosidase” with “α-Glucosidase” as indicated with yellow highlight in the revised manuscript (line 276).

2 Please add the doi number at the very end of the reference.

Response : Thank you for your suggestion. We have added doi  as indicated with the yellow highlight in the revised manuscript.

3 Please indicate in Table 1, Table 3, Table 4 and Table 5 whether there are significant differences between each group. After the analysis, the data were more convincing.

Response : Thank you for your suggestion. We have added statistical analysis (p values) as indicated with the yellow highlight in the revised manuscript.

4 In line 257, there are multiple spaces before "LC-MS/MS".

Response : Thank you for your suggestion.

5 "Tris-CL" in line 218 is changed to "Tris-HCL".

Response : We apologize for this mistake. We have replaced “Tris-CL” with “Tris-HCL” as indicated with yellow highlight in the revised manuscript (line 217)

6 Please unify the upper and lower case of "α-glucosidase" in the whole text. For example, "α-glucosidase in line 37 and 390 is used in the sentence, but" α-Glucosidase "is used in other sentences in the text". 

Response : We apologize for this mistake. We have replaced “α-Glucosidase” with “α-glucosidase” as indicated with yellow highlight in the revised manuscript             (lines29,30,31,33,54,79,81,170,172,178,186,214,268,277,285,288,290,301,318,319,322,324, 335,376,385,387,393,396,399,400,416,432,434,436)

7 Figure 4 and Figure 5 are in reverse order, please correct them.

Response : Thank you for your suggestion.

8 In this article, only the extracted ion current diagram of Mitragynine is presented. It is suggested that the authors add a total ion current diagram for liquid identification and give a brief description of compound identification.

Response : Thank you for your suggestion. We have added a Figure TIC as indicated with yellow highlight in this revised manuscript (Figure 2)

Round 2

Reviewer 2 Report

The authors have revised their manuscript accordingly.